# Exploring High-Order Message-Passing in Graph Transformers

## Abstract

The Transformer architecture has demonstrated promising performance on graph learning tasks. However, the existing attention mechanism used in Graph Transformers (GT) cannot capture high-order correlations that exist in complex graphs, thereby limiting their expressiveness. In this paper, we present a High-Order message-passing strategy within the Transformer architecture (HOtrans) to learn long-range, high-order relationships for graph representation. Recognizing that some nodes share similar properties, we extract communities from the entire graph and introduce a virtual node to connect all nodes in the community. Operating on the community, we adopt a three-step message-passing approach: capture the high-order information of the community into a virtual node; propagate long-range dependent information between communities; aggregate community-level representations back to graph nodes. This facilitates effective global information passing. Virtual nodes capture the high-order community information and support the long-range information passing as the bridge. We demonstrate that many existing GTs can be regarded as special cases of this framework. Our experimental results illustrate that our proposed HOtrans consistently achieves highly competitive results across several node classification tasks.

## 1 Introduction

Learning from graph-structured data, such as social networks, biological networks, and brain networks, is critical for real-world applications. Graph Neural Networks (GNNs) (Kipf & Welling (2017); Veličković et al. (2018); Gasteiger et al. (2019); Hamilton et al. (2017)) have shown promising results on graph representation learning based on a local Message-Passing (MP) scheme, where the information is propagated and aggregated between nodes that are connected in the graph. However, this neighbourhood-dependent information passing strategy limits GNN's capability in achieving long-range dependencies (Zhang et al. (2022)).

Transformer architecture (Vaswani et al. (2017)) which adopts a global attention mechanism has attracted a lot of attention to solve this problem. In contrast to traditional graph neural networks, Graph Transformers (GT) (Kreuzer et al. (2021); Mialon et al. (2021); Ying et al. (2021)) enable information to pass between any two nodes, regardless of the original graph connections. However, these models not only suffer from computational complexity but also face challenges to capture useful topological information (e.g., local high-order correlations of several people in the same club in a social network) of the graph. This is critical for effective graph representation learning. Hence, it is still challenging to effectively achieve long-range dependency while capturing the complex structural relationship in the graph.

In graph learning, hyperedges are introduced to encode complicated correlations by connecting more than two nodes. To this end, the information of multiple nodes can be propagated and aggregated along hyperedges to achieve high-order representation. Inspired by the successful use of patches in the vision domain, some researchers (Gao et al. (2022); Zhao et al. (2023)) have incorporated patch/substructure representations into Graph Transformer. While reducing the computation cost, it showed that introducing high-level representations can benefit graph classification tasks.

In this work, we study a powerful architecture which can effectively propagate information including local-neighbour information, high-order/high-level information, and long-range information. To

address this challenge, we propose a High-Order message-passing scheme within Transformer architecture, which we call to *HOtrans*. To better capture the intricate relationships within a graph, we employ a strategy where we group graph nodes into multiple communities. In each community, all nodes share similar properties (semantic or information). When encoding closer graph nodes into the same community, the challenge is how to capture the local high-order information in the community and propagate it globally for effective and comprehensive representation learning. Consequently, we introduce a virtual node for each community, serving a role akin to hyperedges in hypergraphs, allowing the graph node information to propagate and aggregate along virtual nodes to establish global connections among all nodes.

Based on community-structured data, we adopt a three-step (high-order) message-passing strategy: Graph Node-to-Virtual Node (***G2V*-MP**); Virtual Node-to-Virtual Node (***V2V*-ATTN**); and Virtual Node-to-Graph node (***V2G*-MP**). In the first order, within each community, the information of each node is propagated and aggregated to its corresponding virtual node to capture local high-order information. Then, based on the community-level representations of virtual nodes, we consider a self-attention mechanism between them to allow each virtual node to capture long-range information from another community. Finally, we update the representations of graph nodes by aggregating information from their respective communities. We can see that the virtual nodes effectively connect nearly all nodes in the graph.

Our proposed HOtrans is a general framework and several other existing GT models can be viewed as special cases. If we view the whole graph as a community, our model simplifies to GT models (Wu et al. (2021)) that introduced a special token to connect with all other nodes to achieve global information, which is the lower bound of HOtrans. If we view each node as a community, our model essentially becomes the general Transformer, representing the upper bound of HOtrans. In comparison to these existing GT models, a key advantage of our proposed method is the capability of capturing the higher-order information while saving computation costs (the number of communities is significantly smaller than the number of graph nodes). We evaluate HOtrans on node classification tasks in which GT models have a performance gap. We find significant improvements in accuracy on all datasets including heterophilic datasets and large-scale datasets. In summary, our main contributions are as follows:

- We propose a three-step message-passing framework in Graph Transformer which captures local information, high-order information, and global information to achieve a powerful expressiveness for graph learning.
- We unify the message-passing and Graph Transformer by constructing communities and introducing virtual nodes. We demonstrate that our model is a powerful graph model, i.e., can approximate any other message-passing. We theoretically analyzed the introduced virtual node for each community that enhances the global attention as general Transformers do.
- We conduct extensive experiments on different datasets to demonstrate the effectiveness of the proposed method for node classification. The experimental results provide support the effectivenews of the high-order representation.

## 2 RELATED WORK

### 2.1 GRAPH TRANSFORMERS

Recently, Transformer architecture has been successfully applied to graph domain, showing competitive or even superior performance on many tasks when compared to GNNs. Dwivedi & Bresson (2021) first extended the standard Transformer to graphs with four special designs including position encoding for nodes in a graph. Subsequently, Kreuzer et al. (2021) enabled the position encoding by making it learnable, and further divided the fully connected edges into true edges and virtual edges. There are many other existing GTs (Rong et al. (2020); Zhang et al. (2020); Chen et al. (2021b); Wu et al. (2021); Hussain et al. (2022); Chen et al. (2022a); Nguyen et al. (2022)) and the applications of GTs (Xu et al. (2019); Zhu et al. (2021; 2022); Cai et al. (2022)). A more detailed introduction can also be found in the recent reviews of GTs (Rampášek et al. (2022); Min et al. (2022)). However, the above methods are mostly designed for graph-level tasks due to the time and memory constraints imposed by the self-attention layer. Therefore, several works (Zhao et al. (2021); Choromanski et al.

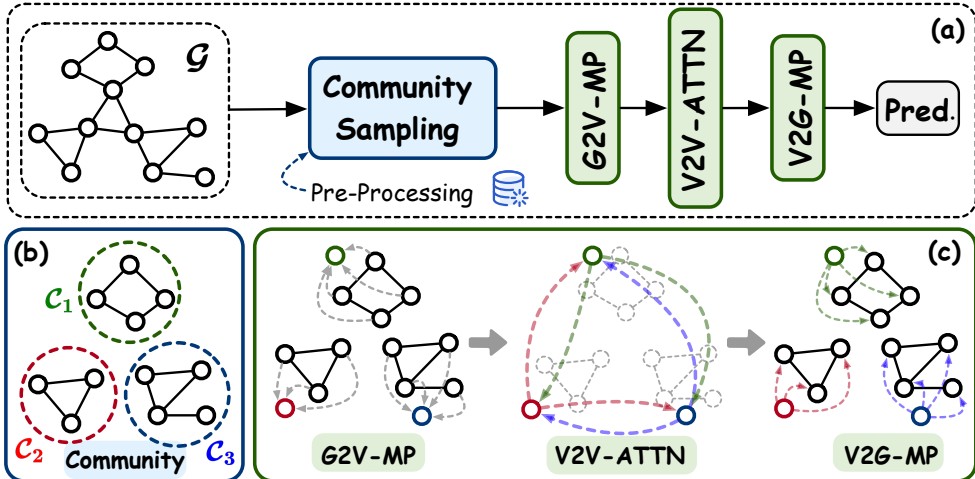

Figure 1: The HOtrans framework. First, adopting community sampling method to obtain multiple communities. Then, propagating and aggregating information in three-step operation: *G2V*-**MP**: Aggregating the high-order information of a community into the virtual node; *V2G*-**MP**: Propagating community-level information in a self-attention mechanism; *V2G*-**MP**: Gathering the updated community-level information for node representation. The upper part of the figure shows the whole pipeline.

(2022); Guo et al. (2022); Park et al. (2022)) have been proposed to make graph transformers more scalable and efficient, but they still suffer from some issues such as missing long-range, high-order information or noise aggregation.

## 2.2 HIGH-ORDER REPRESENTATION LEARNING

In the field of computer vision, it is a common approach to divide the whole image into multiple local patches. Visual Transformers (Dosovitskiy et al. (2020)) then generate the image representation by aggregating high-level representations from these patches rather than individual pixels. Following, Han et al. (2021) introduced a Transformer in Transformer architecture, which further subdivides each local patch into smaller patches. This innovative approach enables the model to capture more detailed representations, thus enhancing feature representations. The high-order, or high-level representations derived from local patches, which often share similar content, play a critical role in learning visual representations. In the graph domain, some works Feng et al. (2019); Wang et al. (2022) also consider encoding high-order correlations for graph representation learning. Typically, the hypergraph structure with a series of hyperedges is introduced to model the complex high-order relationship. Beyond the pairwise connections, hyperedges connect multiple nodes to express their similarities. Within the context of Graph Transformers, Gao et al. (2022); Zhao et al. (2023) have attempted to extract substructures, treating them as patches, and utilizing the substructural representations for graph classification tasks. As graphs continue to grow rapidly in size, the relationships among nodes become increasingly complex. Therefore, exploring and exploiting the high-order representation is significant for graph representation learning.

## 2.3 VIRTUAL NODE IN MESSAGE-PASSING

The introduction of a virtual node expands the graph by adding an extra node that facilitates information exchange among all pairs of nodes. Empirically, its effectiveness in improving performance has been observed in various tasks (Hu et al. (2021b)). Recently, there has been a significant focus on studying its theoretical properties. Hwang et al. (2022) conducted an analysis of the virtual node's role in the context of link prediction. They explored the expressiveness of the learned link representation and its potential impact on under-reaching and over-smoothing. Cai et al. (2023) demonstrated the power of message-passing with a virtual node, showing that it can approximate an arbitrarily self-attention layer within the Graph Transformer architecture.

## 3 METHODOLOGY

In this section, we present our proposed HOtrans framework in Figure 1 to effectively propagate information for comprehensive graph representation. By separating the whole graph into several communities and introducing the virtual node for each community, we achieve the high-order representation of each community and adopt the community-level attention to effectively propagate the high-order long-range dependent information. In the following, we provide the detailed implementation of each component of the architecture. The complexity analysis of HOtrans can be found in the Appendix A.1.

**Notation.** Given an graph $\mathcal{G} = (\mathcal{V}, \mathcal{E})$ with node set $\mathcal{V}$ and edge set $\mathcal{E}$. Suppose there are $N$ nodes in $\mathcal{V}$, the set of edges $\mathcal{E} \subseteq \mathcal{V} \times \mathcal{V}$ define the connections among the $N$ nodes, $(v_i, v_j) \in \mathcal{E}$ denotes the edge between node $v_i$ and node $v_j$. The graph topology is presented by the adjacency matrix $\mathbf{A}$, where $\mathbf{A}_{ij} = 1$ if there exists an edge $(v_i, v_j)$, $\mathbf{A}_{ij} = 0$ otherwise. We denote $\mathbf{X} \in \mathbb{R}^{N \times d}$ the node features, where each node $i$ has $x \in \mathbb{R}^d$. Let $y_i$ denote the label of node $i$, in this work, we focus on the node classification task which aims to predict the labels of the unknown nodes in the graph.

**Transformer architecture.** The Transformer architecture consists of a composition of Transformer layers. Each Transformer layer has a self-attention module and a position-wise feed-forward network (FFN). The self-attention mechanism calculates attention scores by taking the inner product of query vectors ($\mathbf{Q}$) and key vectors ($\mathbf{K}$). It then uses these scores to aggregate value vectors ($\mathbf{V}$) in a weighted manner, resulting in contextualized representations, that is,

$$\mathbf{Q} = \mathbf{H}\mathbf{W}^Q, \mathbf{K} = \mathbf{H}\mathbf{W}^K, \mathbf{V} = \mathbf{H}\mathbf{W}^V; \tag{1}$$

$$\mathbf{A} = \frac{\mathbf{Q}\mathbf{K}^\top}{\sqrt{d'}}, \quad \mathrm{Attn}(\mathbf{H}) = \mathrm{softmax}(\mathbf{A})\mathbf{V} \tag{2}$$

where $\mathbf{W}^Q \in \mathbb{R}^{d \times d'}, \mathbf{W}^K \in \mathbb{R}^{d \times d'}$, and $\mathbf{W}^V \in \mathbb{R}^{d \times d'}$ are projection matrices, $\mathbf{H} = \left[\boldsymbol{h}_1^\top, \ldots, \boldsymbol{h}_n^\top\right] \in \mathbb{R}^{n \times d}$ denotes the input matrix of node embeddings, and $d'$ is the output hidden dimension. Generally, it is the global attention mechanism that allows everything to connect to everything. Instead of performing a single attention function, it is standard to adopt multi-head attention (MHA).

### 3.1 COMMUNITY SAMPLING

Effectively utilising the structural information of the graph is the key challenge for graph representation learning. We note that data correlations in practice are usually beyond pairwise and could be more complex. Typically, a community of friends share their common interest in basketball in a social network. To encode these high-order correlations, we consider extracting meaningful communities from the whole graph. Here, a community is introduced to collect multiple vertices sharing similar properties (semantic or information), similar to how a hyperedge connects multiple objects in graph learning. Considering the real-world applications, we apply two approaches: random walk (Zeng et al. (2019)) and spectral clustering (Chiang et al. (2019)). The details of two sampling methods can be found in A.2.

### 3.2 MODEL DESIGN

Operating on communities, our proposed method involves three steps to obtain high-order long-range information: Graph Node-to-Virtual Node: gathering high-order information of a community into the virtual node; Virtual Node-to-Virtual Node: propagating the high-order information globally; and Virtual Node-to-Graph Node: achieving high-order long-range dependent information for node representation.

**Graph Node-to-Virtual Node (*G2V*-MP).** To capture the high-order information in the community, we introduce a virtual node (VN) for each community, and connect it with other nodes in the community. The virtual node trick, the use of an additional node that connects to all input graph nodes, has been observed to improve GNNs (Gilmer et al. (2017); Hu et al. (2021a); Wu et al. (2021)) and has been demonstrated theoretically (Cai et al. (2023)). Instead of acting like the READOUT aggregating whole information in the graph, we introduce the virtual node for each community, expecting to capture the high-order structural information of a graph for effective Transformer.

Assume there are $m$ communities $\left\{\tilde{\mathcal{V}}_1, \ldots, \tilde{\mathcal{V}}_m\right\}$, thus, we have virtual nodes $\overline{V} = \{\overline{v}_1, \ldots, \overline{v}_m\}$. We initialize the virtual node feature $\overline{x}_i$ with $d$-dimensional zero vector. Note that the number of virtual nodes is significantly smaller than the number of graph nodes. For each community $\tilde{\mathcal{V}}_i$, the community representation can be obtained by the virtual node acting as the query $\overline{q}_i$ with $\overline{q}_i = \mathbf{W}^Q \overline{x}_i$:

$$\boldsymbol{h}_i^c = \text{softmax}\left(\alpha \overline{\boldsymbol{q}}_i^T \tilde{\mathbf{K}}_{\mathcal{V}_i}\right) \tilde{\mathbf{V}}_{\mathcal{V}_i}^T, \tag{3}$$

where $\alpha$ is a constant scalar ($\alpha = \frac{1}{\sqrt{d'}}$), $\tilde{\mathbf{K}}_{\mathcal{V}_i}$ and $\tilde{\mathbf{V}}_{\mathcal{V}_i}$ are the key and value matrices of $\overline{v}_i$'s community. To this end, the virtual node aggregate the community-level information.

**Virtual Node-to-Virtual Node (*V2V*-ATTN).** To maintain the benefit of global attention in Transformer architecture, we enable information propagation between any two communities. Viewing each virtual node as a token, we adopt self-attention to refine the community-level representations:

$$\text{Attn}(\mathbf{H}^c) = \text{softmax}(\frac{\mathbf{Q}^c \mathbf{K}^{c\top}}{\sqrt{d'}})\mathbf{V}^c, \tag{4}$$

where $\mathbf{Q}^c = \mathbf{H}^c \mathbf{W}^Q$, $\mathbf{K}^c = \mathbf{H}^c \mathbf{W}^K$, $\mathbf{V}^c = \mathbf{H}^c \mathbf{W}^V$, with $\mathbf{H}^c = \left[\boldsymbol{h}_1^{c\top}, \ldots, \boldsymbol{h}_n^{c\top}\right]$. By propagating information between communities, we obtain the updated community representation $\mathbf{H}^{c'}$. The information passing from community to community helps to: (1) Enhance the relationship of communities; and (2) Capture the long-range dependency at community-level.

**Virtual Node-to-Graph Node (*V2G*-MP).** To finally obtain the representation of each node, we aggregate the community representation to update node features.

We define the query vector of graph node $v_i$ as $\boldsymbol{q}_i$, while the key and value matrices from introduced virtual nodes are $\mathbf{K}^{c'} \in \mathbb{R}^{m \times d'}$ and $\mathbf{V}^{c'} \in \mathbb{R}^{m \times d'}$, respectively.

For graph node $v_i$, its representation can be enhanced with community-level representations as:

$$\boldsymbol{h}_i = \text{softmax}\left(\alpha \boldsymbol{q}_i^T \mathbf{K}_{V(i)}^{c'}\right) \mathbf{V}_{V(i)}^{c'}{}^T, \tag{5}$$

where $\mathbf{K}_{V(i)}^{c'}$ and $\mathbf{V}_{V(i)}^{c'}$ are the key and value matrices of $v_i$'s communities.

Considering the importance of neighbours, it is also necessary to maintain local message-passing (Zhao et al. (2021)) for the local-dependency graph data. Thus, the representation of graph node $v_i$ can be updated as follows:

$$\boldsymbol{h}_i = \text{softmax}\left(\alpha \boldsymbol{q}_i^T \mathbf{K}_{V(i)}\right) \mathbf{V}_{V(i)}^T, \tag{6}$$

where $\mathbf{K}_{V(i)}$ is the combination of $\mathbf{K}_{V(i)}^{c'}$ and $\mathbf{K}_{\mathcal{N}(i)}$, and $\mathbf{V}_{V(i)}$ is the combination of $\mathbf{V}_{V(i)}^{c'}$ and $\mathbf{V}_{\mathcal{N}(i)}]$, where $\mathbf{K}_{\mathcal{N}(i)}, \mathbf{V}_{\mathcal{N}(i)}$ are the key and value matrices of neighboring nodes of $v_i$, respectively.

**Implementation Details in HOtrans.** We have presented a single attention mechanism in line with general Transformers. In practice, a multi-head attention following with feed-forward blocks with layer normalization ($\text{LN}(\cdot)$) is adopted in our proposed framework as:

$$\boldsymbol{h}'^{(l)} = \text{LN}\left(\text{MHA}\left(\boldsymbol{h}^{(l-1)}\right)\right) + \boldsymbol{h}^{(l-1)}; \boldsymbol{h}^{(l)} = \text{LN}\left(\text{FNN}\left(\boldsymbol{h}'^{(l)}\right)\right) + \boldsymbol{h}'^{(l)}. \tag{7}$$

The positional encoding is an important component in Transformer, in the graph domain, researchers integrated the positional information into Graph Transformers by random walk positional encoding (Dwivedi & Bresson (2021)), or Laplacian positional encoding (Dwivedi et al. (2021)). In our method, we also consider these positional encodings, and test their performance in Appendix A.6.

## 4 THEORETICAL ANALYSIS

In this section, We analyze several cases of HOtrans: the lower bound of HOtrans, the upper bound of HOtrans, and the general Hotrans, and demonstrate that HOtrans is a powerful model which

can approximate GT model to achieve global attention, i.e., unifying MP and GT with community and virtual nodes. We further analyzed the role of virtual nodes in capturing the high-order representation in HOtrans versus the function of hyperedges in hypergraph convolutional networks in Appendix A.3.

**View the whole graph as a community.** Consider the whole graph as a community, the GT can be simplified by Message-Passing Neural Networks (MPNN) with a virtual node that connects to all graph nodes, we refer this to the lower bound of HOtrans. It has been demonstrated in (Cai et al. (2023)), MPNN with a virtual node can approximate a self-attention layer arbitrarily well.

**View each node as a community.** By viewing each node as a community, the proposed HOtrans is actually the standard Transformer. Specifically, the three-step message-passing in HOtrans is reduced to one step: Virtual Node-to-Virtual Node. When the community just contains one node, the virtual node is itself. To this end, HOtrans propagates information between any two nodes.

**Multiple communities with multiple nodes.** In the general case there are multiple communities with each one containing multiple nodes. Here, we demonstrate the powerful of HOtrans in the general case by showing the information passing with virtual nodes can approximate a global-attention arbitrarily well.

**Definition 4.1.** A full self-attention layer is defined as the following form:

$$
\begin{aligned}
\boldsymbol{x}_i^{(l+1)} &= \sum_{j=1}^n \frac{\phi\left(\boldsymbol{q}_i\right)^T \phi\left(\boldsymbol{k}_j\right)}{\sum_{k=1}^n \phi\left(\boldsymbol{q}_i\right)^T \phi\left(\boldsymbol{k}_k\right)} \cdot \boldsymbol{v}_j \\
&= \frac{\left(\phi\left(\boldsymbol{q}_i\right)^T \sum_{j=1}^n \phi\left(\boldsymbol{k}_j\right) \otimes \boldsymbol{v}_j\right)^T}{\phi\left(\boldsymbol{q}_i\right)^T \sum_{k=1}^n \phi\left(\boldsymbol{k}_k\right)},
\end{aligned}
\tag{8}
$$

where $\phi(\cdot)$ is a low-dimensional feature map with random transformation, $\boldsymbol{q}_i$, $\boldsymbol{k}_i$, $\boldsymbol{v}_i$ are the query, key, and value vector, respectively.

**Proposition 4.1.** *The $\sum_{k=1}^n \phi\left(\boldsymbol{k}_k\right)$ and $\sum_{j=1}^n \phi\left(\boldsymbol{k}_j\right) \otimes \boldsymbol{v}_j$ can be approximated by the virtual node, and shared for all graph nodes, using only $\mathcal{O}(1)$ layers of MPNNs.*

**Theorem 4.1.** *Message-Passing with virtual node for the community, following with a self-attention between virtual node and another Message-Passing with virtual node for graph node can approximate self-attention arbitrarily well.*

## 5 EXPERIMENTS

In this section, we evaluate the effectiveness of HOtrans for graph node classification, where GT models have yet to demonstrate state of the art performance. We compare HOtrans with a series of baseline models including standard GCN-based models, hypergraph-based models, heterophilic-graph-oriented models, and Transformer-based models. Then, we ablate the important design elements of the proposed HOtrans including the number of community, the necessarity of message-passing between communities, the local connections in graph, and positional encoding. The detailed experiment seeting can be found in Appendix A.5.

**Datasets.** We conducted experiments on a wide range of graph benchmarks: 1) homophilic graph datasets (Cora, Citeseer, Pubmed, DBLP, CoraFull, and ogbn-arxiv) and 2) heterophilic graph datasets (Cornell, Texas, Wisconsin, and Actor), involving diverse domains and sizes (ogbn-arxiv is a large-scale dataset). The details of the datasets are provided in Appendix 4.

### 5.1 COMPARISON TO THE STATE-OF-THE-ART

**Performance on Homophilic Graphs.** The homophilic datasets are the graphs with high **Homo.** (indicating the proportion of edges connecting nodes with the same label (Zhu et al. (2020))). Focus on prediction accuracies of node classification, we report the results in Table 1. From the table, we can observe that the proposed HOtrans, regardless of the sampling methods, achieves the state-of-the-art, or competitive performance on all datasets.

Compared with GCN-based methods, HOtrans performs better on graphs with more nodes (*e.g.*, Pubmed), specifically, HOtrans improves upon the state-of-that art GNN method-APPNP and HGNN by a margin of 2.7% and 1.3% on Pubmed and Cora, respectively. This is likely because, based on local message-passing, GCN methods have the disadvantage of capturing long-range dependencies. In contrast, our HOtrans enables the learning of more informative node representations, including community-level and global-level information, which represents a significant advantage, especially for larger graphs that are more complex.

We can find that in comparison to Graph Transformer-based methods, we can see the obvious advantage of HOtrans on the small-scale datasets (*e.g.*, Cora and Citeseer) with higher **Homo.**, i.e., local-neighbourhood information is more important. Thus, the vanilla global attention on the whole graph adopted in existing GTs (such as Graphormer) leads to massive unrelated information aggregation. By introducing the concept of community and virtual node, our proposed method incorporates local high-order information from the community and global information from community-attention can better encode the correlations in complicated graphs.

Our proposed method can be generalized to large-scale graphs, ogbn-arxiv, while some other GT methods cannot be applied to such graphs due to their poor scalability. We have noticed that Graphformer and LiteGT encounter out-of-memory errors, even when processing small graphs. This highlights the need for a Graph Transformer that can scale effectively to handle large-scale graphs.

**Performance on Heterophilic Graphs.** We further evaluate the effectiveness of the proposed method on heterophilic graphs. These heterophilic datasets are usually small-scale but low **Homo.**, thus, can be viewed as long-range dependency datasets. From the results in Table 2, we can observe that the heterophily-based methods that are specially designed for these datasets can generally achieve better performance than other GCN-based methods. Except for Gapformer, most of the existing Transformer-based models also show poor performance, which implies that GTs fail to propagate and aggregate useful information. Instead, our proposed method can be extended to heterophilic graph datasets and achieves improved performance. Note that in Gapformer, the global information is aggregated by the pooling method, which can be viewed as our HOtrans in the case that considers the whole graph as a community.

**Evaluation on community sampling methods.** Focusing on sampling methods, we can observe that HOtrans(randomwalk) with random walk sampling slightly outperforms HOtrans(clustering) with spectral clustering method. Note the differences of random walk sampling with spectral clustering: 1) with a small number communities, the obtained communities using random walk cannot contain all graph nodes while spectral clustering method put everynode into a community. 2) the random walk sampling constraint the nodes in a community in $k$-hop walk length, thus, it contain more local structural information while spectral clustering seperate the graph from a global view. Thus, random walk sampling capture more local information than spectral clustering methods.

The improved performance of HOtrans in various datasets validates the significance of comprehensively considering the complicated correlations in the graph and exploring and exploiting a high-order message-passing approach to effectively propagate information in graph-structured data.

## 5.2 ABLATION STUDIES

We perform ablation studies to verify how different configurations of our model can affect its performance. The effect of positional encodings can be found in Appendix A.6.

### 5.2.1 EFFECT OF THE NUMBER OF COMMUNITY.

Community extraction is an important component in our proposed HOtrans. As discussed earlier, according to the number of community, the lower bound of HOtrans considers the entire graph as a single community. In this way, HOtrans can be simplified to the MPNNs which introduces a virtual node to encode global information. If constructing a community for each node, we get the upper bound of HOtrans, similar to a vanilla Transformers that connect any two nodes.

We analyze the effect of the number of communities with two sampling methods for HOtrans. From the results in Figure 2, we can find that increasing the number of community in the early stage can enhance the performance of HOtrans on Cora. This is because that HOtrans encode more local

Table 1: Experimental results for the node classification task on different datasets (mean accuracy (%) and standard deviation over 10 different runs). Red: the best performance per dataset. Blue: the second best performance per dataset. OOM denotes out-of-memory.

| | Cora | Citeseer | Pubmed | DBLP | CoraFull | ogbn-arxiv |
|---|---|---|---|---|---|---|
| *GCN-based methods* | | | | | | |
| GCN Kipf & Welling (2017) | $86.92_{\pm 1.33}$ | $76.13_{\pm 1.51}$ | $87.01_{\pm 0.62}$ | $85.13_{\pm 0.44}$ | $24.49_{\pm 0.47}$ | $70.40_{\pm 0.10}$ |
| APPNP Gasteiger et al. (2019) | $87.75_{\pm 1.30}$ | **$76.53_{\pm 1.61}$** | $86.52_{\pm 0.61}$ | $85.22_{\pm 0.56}$ | $20.61_{\pm 0.78}$ | $70.20_{\pm 0.16}$ |
| GCNII Chen et al. (2020) | $86.08_{\pm 2.18}$ | $74.75_{\pm 1.76}$ | $85.98_{\pm 0.61}$ | $83.26_{\pm 0.49}$ | $9.10_{\pm 0.62}$ | $69.78_{\pm 0.16}$ |
| GAT Veličković et al. (2018) | $87.34_{\pm 1.14}$ | $75.75_{\pm 1.86}$ | $85.37_{\pm 0.56}$ | $83.86_{\pm 0.44}$ | $25.32_{\pm 1.43}$ | $67.56_{\pm 0.12}$ |
| GATv2 Brody et al. (2022) | $87.25_{\pm 0.89}$ | $75.72_{\pm 1.30}$ | $85.75_{\pm 0.55}$ | $84.96_{\pm 0.47}$ | $31.62_{\pm 0.71}$ | $68.84_{\pm 0.13}$ |
| HGNN Feng et al. (2019) | $86.88_{\pm 1.22}$ | $75.87_{\pm 1.47}$ | - | - | - | OOM |
| *Graph Transformer-based methods* | | | | | | |
| SAN Kreuzer et al. (2021) | $81.91_{\pm 3.42}$ | $69.63_{\pm 3.76}$ | $81.79_{\pm 0.98}$ | – | $45.61_{\pm 5.25}$ | $69.17_{\pm 0.15}$ |
| Graphormer Ying et al. (2021) | $67.71_{\pm 0.78}$ | $73.30_{\pm 1.21}$ | OOM | OOM | OOM | OOM |
| LiteGT Chen et al. (2021a) | $80.62_{\pm 2.69}$ | $69.09_{\pm 2.03}$ | $85.45_{\pm 0.69}$ | – | $56.86_{\pm 0.69}$ | OOM |
| UniMP Shi et al. (2020) | $84.18_{\pm 1.39}$ | $75.00_{\pm 1.59}$ | $88.56_{\pm 0.32}$ | $84.25_{\pm 0.42}$ | $67.93_{\pm 0.56}$ | **$73.19_{\pm 0.18}$** |
| ANS-GT Zhang et al. (2022) | $86.71_{\pm 1.45}$ | $74.57_{\pm 1.51}$ | **$89.76_{\pm 0.46}$** | $85.19_{\pm 0.47}$ | $61.66_{\pm 1.85}$ | – |
| Gapformer | $87.37_{\pm 0.76}$ | $76.21_{\pm 1.47}$ | $88.98_{\pm 0.46}$ | **$85.50_{\pm 0.43}$** | **$68.22_{\pm 0.70}$** | **$71.90_{\pm 0.19}$** |
| HOtrans (randomwalk) | **$88.11_{\pm 1.05}$** | **$76.74_{\pm 1.47}$** | **$89.20_{\pm 1.34}$** | **$85.58_{\pm 0.24}$** | **$68.95_{\pm 0.65}$** | $70.67_{\pm 0.14}$ |
| HOtrans (clustering) | **$88.09_{\pm 1.34}$** | $76.35_{\pm 1.47}$ | $88.96_{\pm 0.49}$ | $85.26_{\pm 0.36}$ | $66.76_{\pm 0.46}$ | $69.89_{\pm 0.16}$ |

Figure 2: The ablation study on the number of communities. We set the number of communities to 1 (the whole graph as a community) and $1\%, 10\%, 20\%, 50\%$ of the number of graph nodes.

high-order information with more communities extracted by random walk method. As the number of communities increases, we can observe a decrease trend followed by an increase for HOtrans with spectral clustering method on Cora. This shows that there exists some important substructures in the graph. We can see the stable performance of HOtrans on Wisconsin with different numbers of communities for both methods. While Wisconsin is a small-scale dataset, the global information can be well encoded by introducing a virtual node.

### 5.2.2 ABLATION OF TRANSFORMER COMPONENTS

We perform a series of ablation studies to test the importance of some designs in our proposed HOtrans. We report the results in Table 5.

**Effect of message-passing between communities.** As we analyzed in A.3, dropping out the second step (*V2V*-**ATTN**), our HOtrans is similar to popular hypergraph-based neural networks from the perspective of message-passing. We can observe that when not taking into account the relationships between communities, HOtrans (*V2V*-**ATTN**)) exhibits a significant performance degradation compared to HOtrans. Without *V2V*-**ATTN**, the node representation is still limited in the local neighbourhood, i.e., community. Propagating information between communities can help the node finally capture the high-order long-range dependency in the whole graph.

**Effect of local information for different datasets.** While the one major advantage of Transformer is capturing the long-range dependency in objects, we examine the importance of local information for some datasets. We can observe that it can improve the performance if we consider the local

Table 2: Experimental results for the node classification task on four heterophilic datasets (mean accuracy (%) and standard deviation over 10 different runs). Red: the best performance per dataset. Blue: the second best performance per dataset.

| | Cornell | Texas | Wisconsin | Actor |
|---|---|---|---|---|
| *GCN-based methods* | | | | |
| GCN Kipf & Welling (2017) | $45.67_{\pm 7.96}$ | $60.81_{\pm 8.03}$ | $52.55_{\pm 4.27}$ | $28.73_{\pm 1.17}$ |
| APPNP Gasteiger et al. (2019) | $41.35_{\pm 7.15}$ | $61.62_{\pm 5.37}$ | $55.29_{\pm 3.90}$ | $29.42_{\pm 0.81}$ |
| GAT Veličković et al. (2018) | $47.02_{\pm 7.66}$ | $62.16_{\pm 4.52}$ | $57.45_{\pm 3.51}$ | $28.33_{\pm 1.13}$ |
| GATv2 Brody et al. (2022) | $50.27_{\pm 8.97}$ | $60.54_{\pm 4.55}$ | $52.74_{\pm 3.96}$ | $28.79_{\pm 1.47}$ |
| *Heterophily-based methods* | | | | |
| MLP LeCun et al. (2015) | $71.62_{\pm 5.57}$ | $77.83_{\pm 5.24}$ | $82.15_{\pm 6.93}$ | $33.26_{\pm 0.91}$ |
| MixHop Abu-El-Haija et al. (2019) | $76.48_{\pm 2.97}$ | $83.24_{\pm 4.48}$ | $85.48_{\pm 3.06}$ | $34.92_{\pm 0.91}$ |
| H2GCN Zhu et al. (2020) | $75.40_{\pm 4.09}$ | $79.73_{\pm 3.25}$ | $77.57_{\pm 4.11}$ | $36.18_{\pm 0.45}$ |
| FAGCN Bo et al. (2021) | $67.56_{\pm 5.26}$ | $75.67_{\pm 4.68}$ | $75.29_{\pm 3.06}$ | $32.13_{\pm 1.33}$ |
| GPRGNN Chien et al. (2021) | $76.76_{\pm 2.16}$ | $81.08_{\pm 4.35}$ | $82.66_{\pm 5.62}$ | $35.30_{\pm 0.80}$ |
| *Graph Transformer-based methods* | | | | |
| SAN Kreuzer et al. (2021) | $50.85_{\pm 8.54}$ | $60.17_{\pm 6.66}$ | $51.37_{\pm 3.08}$ | $27.12_{\pm 2.59}$ |
| UniMP Shi et al. (2020) | $66.48_{\pm 12.5}$ | $73.51_{\pm 8.44}$ | $79.60_{\pm 5.41}$ | $35.15_{\pm 0.84}$ |
| NAGphormer Chen et al. (2022b) | $56.22_{\pm 8.08}$ | $63.51_{\pm 6.53}$ | $62.55_{\pm 6.22}$ | $34.33_{\pm 0.94}$ |
| Gapformer | $77.57_{\pm 3.43}$ | $80.27_{\pm 4.01}$ | $83.53_{\pm 3.42}$ | $36.90_{\pm 0.82}$ |
| HOtrans (randomwalk) | $\textcolor{red}{79.46}_{\pm 2.16}$ | $\textcolor{red}{83.44}_{\pm 1.87}$ | $\textcolor{red}{87.25}_{\pm 2.67}$ | $\textcolor{red}{38.11}_{\pm 0.87}$ |
| HOtrans (clustering) | $\textcolor{blue}{78.65}_{\pm 2.82}$ | $\textcolor{blue}{82.63}_{\pm 4.97}$ | $\textcolor{blue}{86.47}_{\pm 2.97}$ | $\textcolor{blue}{37.44}_{\pm 0.68}$ |

Table 3: Abalation study on different datasets.

| Community Sampling | Model | Cora | Citeseer | Cornell | Texas | Wisconsin |
|---|---|---|---|---|---|---|
| Random Walk | HOtrans(w/o *V2V*-**ATTN**)) | 83.14±1.48 | 74.94±1.64 | 77.57±3.21 | 80.54±3.59 | 85.89±2.60 |
| | HOtrans | 88.11±1.05 | 76.74±1.47 | 76.49±2.72 | 80.00±4.22 | 87.25±2.67 |
| Random Walk | HOtrans(w/o local) | 83.04±1.48 | 74.47±2.10 | 76.49±2.72 | 82.70±4.86 | 83.44±1.87 |
| | HOtrans(w local) | 88.11±1.05 | 76.74±1.47 | 70.27±2.34 | 74.90±2.78 | 78.19±2.67 |

neighbours in the third step (*G2V*-**MP**) for Cora and Citeseer as they are small-scale datasets with high Homo. In contrast, it is more beneficial to disregard the original graph connections for Cornell, Texas, and Wisconsin.

## 6 CONCLUSION

In this paper, we introduce a high-order message-passing strategy within the Transformer architecture to learn long-range, high-order relationships for graph representation. Initially, we extract communities from the entire graph and introduce a virtual node for each community. Subsequently, leveraging community-structured data, we adopt a three-step message-passing scheme to aggregate information from graph node to virtual node, propagate information between virtual nodes which represent the high-order information of communities, and send the community-level information in virtual nodes back to graph nodes. The introduced virtual nodes act like hyperedge in hypergraph to effectively propagate information to other graph nodes. We theoretically demonstrate the powerful expressiveness of HOtrans and empirically show the effectiveness of HOtrans across diverse datasets on node classification. In this paper, we just evaluate HOtrans on node classification, we will test the proposed method on more other graph tasks in the future.

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

# A    APPENDIX

## A.1    COMPLEXISTY ANALYSIS OF HOTRANS

We analyze the complexity of HOtrans. The computational complexity of the first step Graph Node-to-Virtual Node is $\mathcal{O}\left(mN\right)$. Since $m$ is the number of community and usually much smaller than the number of graph nodes $N$, the computational complexity can be simplified as $\mathcal{O}(N)$. Moreover, the computational complexity of the second step Virtual Node-to-Virtual Node is $\mathcal{O}\left(m^2\right)$, it is a self-attention. The final step Virtual Node-to-Virtual Node is $\mathcal{O}\left(mN\right)$. Therefore, the overall complexity of Gapformer is $\mathcal{O}(m^2 + N)$.

## A.2    THE COMMUNITY SAMPLING METHODS

**Random walk sampling.** To preserve the graph structural information as well as local or long-range connectivity, random walk sampling is a simple but effective approach. We consider a regular random walk sampler with $m$ root nodes selected uniformly at random and each walker goes $k$ hops. As such, we can obtain the communities $\left\{\tilde{\mathcal{V}}_1, \ldots, \tilde{\mathcal{V}}_m\right\}$. Each community $\tilde{\mathcal{V}}_i$ has $k+1$ nodes which are $k$-hop neighbours.

**Spectral clustering.** Spectral clustering methods segment the graph by minimum cuts such that the number of within-cluster links is much higher than between-cluster links in order to better capture good community structure. However, these spectral clustering methods can just obtain non-overlapping clusters. As we aim to achieve more communication between communities, we extend each cluster with its 1-hop neighbourhood He et al. (2023). Thus, we can obtain $m$ communities $\left\{\tilde{\mathcal{V}}_1, \ldots, \tilde{\mathcal{V}}_m\right\}$, where $\tilde{\mathcal{V}}_i \leftarrow \tilde{\mathcal{V}}_i \cup \left\{\mathcal{N}_1(j) \mid j \in \tilde{\mathcal{V}}_i\right\}$.

## A.3    CONNECTION BETWEEN VIRTUAL NODE AND HYPEREDGE

We analyze the role of virtual nodes in capturing the high-order representation in HOtrans versus the function of hyperedges in hypergraph convolutional networks.

**Encode complex relationship.** To encode the high-order correlations in the complicated graph, in hypergraph convolutional networks (HGCN), the hyperedges are introduced to connect multiple nodes. In this work, we introduce a virtual node for each community which contains multiple nodes

sharing similar properties (semantic or information). Like the hyperedge, the virtual node connects with every node in its community.

**High-Order Message-Passing.** Following the message-passing scheme, HGCN first propagates and aggregates information along hyperedge $e^h$ to obtain the hyperedge presentation $\boldsymbol{a}_{e^h}$, then updates the node representation by aggregating the hyperedge representations. Formally, the layer-wise message-passing is defined as:

$$\boldsymbol{a}_{e^h}^{(k)} = \text{Aggregate}^{(k)} \left( \left\{ \boldsymbol{z}_u^{(k-1)} : u \in e^h \right\} \right), \boldsymbol{z}_v^{(k)} = \text{Update}^{(k)} \left( \left\{ \boldsymbol{a}_{e^h}^{(k)} : v \in e \right\} \right), \quad (9)$$

where $\boldsymbol{z}_v^{(k)}$ is the feature vector of node $v$ at the $k^{th}$ layer. The hypergraph-based convomutional networks design $\text{Aggregate}^{(k)}(\cdot)$ and $\text{Combine}^{(k)}(\cdot)$ operations based on hypergraph structure.

For example, in a spectral-based hypergraph convolutional network, the convolutional operation is defined as:

$$\boldsymbol{\Delta} = \mathbf{D}_v^{-1/2} \mathbf{S} \mathbf{W} \mathbf{D}_e^{-1} \mathbf{S}^\top \mathbf{D}_v^{-1/2}, \boldsymbol{h}^{(k)} = \sigma \left( \boldsymbol{\Delta} \boldsymbol{z}^{(k-1)} \theta^{(k)} \right), \quad (10)$$

where the diagonal matrices $\mathbf{D}_v$ and $\mathbf{D}_e$ denote the vertex and hyperedge degrees, respectively. $\mathbf{W}$ indicate the relationship of hyperedges, the incidence matrix $\mathbf{S}$ denote the correlations of nodes and hyperedges with $S(v, e) = \begin{cases} 1, & \text{if } v \in e \\ 0, & \text{if } v \notin e \end{cases}$, $\theta^k$ is the weights of $k^{th}$ layer. Based on the hyperedge operation, we can refine the message-passing in Eq. 10 into three steps: node-to-hyperedge, hyperedge-to-hyperedge, hyperedge-to-node with the approximate presentation:

$$\boldsymbol{a}_{e^h}^{(k)} = \mathbf{S}^\top \boldsymbol{z}^{(k-1)}, \boldsymbol{a}_{e^h}^{(k)} = \mathbf{W} \boldsymbol{a}_{e^h}^{(k)}, \boldsymbol{z}^{(k)} = \mathbf{S} \boldsymbol{a}_{e^h}^{(k)}. \quad (11)$$

We can see that the three-step message-passing in HGCN is equivalent to the three-step operation in HOtrans. In HGCN, the relationship of hyperedges usually can be ignored, i.e., $\mathbf{W} = \mathbf{I}$. In HOtrans, the framework can also be simplified to two steps without Virtual Node-to-Virtual Node. From a high level, graph convolutional neural networks can be viewed as special cases of hypergraph convolutional networks. In comparison, our proposed HOtrans framework can be simplified to other existing GT models.

## A.4 PROOF

Based on **Proposition 4.1**, in the process of Graph Node-to-Virtual Node (**G2V-MP**), the message-passing with a virtual node in a community is powerful to update virtual node as:

$$\tau_{j \in [n]} \phi_{\boldsymbol{G2V-MP}}^{(k)} \left( \cdot, \{\boldsymbol{x}_i\}_i \right) = \left[ \sum_{j=1}^n \phi(\boldsymbol{k}_j), \ f \left( \sum_{i=1}^n \phi(\boldsymbol{k}_j) \otimes \boldsymbol{v}_j \right) \right], \quad (12)$$

where $f(\cdot)$ flattens a 2D matrix to a 1D vector in raster order.

Then, in the process of Virtual Node-to-Virtual Node (**V2V-ATTN**), a self-attention mechanism ($\gamma_{\boldsymbol{V2V-ATTN}}$) is adopted to propagate information between any two virtual nodes. The updated virtual nodes can be represented as:

$$\overline{\boldsymbol{h}}_i = \gamma_{\boldsymbol{V2V-ATTN}} \left( \left[ \sum_{j=1}^n \phi(\boldsymbol{k}_j), \ f \left( \sum_{i=1}^n \phi(\boldsymbol{k}_j) \otimes \boldsymbol{v}_j \right) \right] \right) \quad (13)$$

Finally, the updated virtual node sends its message back to all other nodes in Virtal Node-to-Graph Node (**V2G-MP**). Each graph node $v_i$ applies the function:

$$g_{\boldsymbol{V2G-MP}} \left( \boldsymbol{x}_i, \overline{\boldsymbol{h}}_i \right) = \frac{\left( \phi(\boldsymbol{q}_i) \sum_{j=1}^n \phi(\boldsymbol{k}_j) \otimes \boldsymbol{v}_j \right)^T}{\phi(\boldsymbol{q}_i)^T \sum_{k=1}^n \phi(\boldsymbol{k}_k)} \quad (14)$$

Therefore, the information of a graph node can be propagated to any other nodes by the virtual nodes as the bridges.

Table 4: Statistics of benchmark datasets.

|  | Cora | Citeseer | Pubmed | DBLP | CoraFull | ogbn-arxiv | Cornell | Texas | Wisconsin | Actor |
|---|---|---|---|---|---|---|---|---|---|---|
| **# Nodes** | 2,708 | 3,327 | 19,717 | 17,716 | 19,793 | 169,343 | 183 | 183 | 251 | 7,600 |
| **# Edges** | 5,429 | 4,732 | 44,338 | 105,734 | 126,842 | 1,166,343 | 280 | 195 | 466 | 26,752 |
| **Homo.** | 0.83 | 0.72 | 0.79 | 0.70 | 0.57 | 0.63 | 0.30 | 0.11 | 0.21 | 0.22 |

Table 5: Abalation study of positional encoding on different datasets.

| Community Sampling | Model | Cora | Citeseer | Cornell | Texas | Wisconsin |
|---|---|---|---|---|---|---|
| Spectral Clustering | HOtrans(lpe) | 86.24±1.33 | 75.87±1.75 | 71.35±4.05 | 77.30±7.37 | 81.96±3.26 |
|  | HOtrans(rwpe) | 86.22±1.53 | 75.65±1.78 | 73.78±3.83 | 78.38±4.01 | 84.71±2.11 |
|  | HOtrans(w/o pe) | 86.32±1.33 | 76.06±1.68 | 78.65±2.82 | 80.54±4.80 | 86.08±2.97 |

## A.5 EXPERIMENTAL PART

**Settings.** For Cora, Citeseer, and Pubmed datasets, we follow the same experimental procedure, such as features and data splits in Pei et al. (2020). For heterophilic graph datasets, we adopt the same dataset splits used by Zhu et al. (2020). For other datasets, we randomly split them into 60%/20%/20% as training/validation/test sets following Zhang et al. (2022); Liu et al. (2023). We adopt two sampling methods-random walk Zeng et al. (2019) and spectral clustering Chiang et al. (2019) to extract communities. We set the number of communities to 1 (the whole graph as a community) and 1%, 10%, 20%, 50% of the number of nodes in the graph. The training utilizes Adam optimizer Kingma & Ba (2014) for GNN methods, while Adamw is adopted for all Graph Transformer-based models. Each method runs for 200 epochs on all datasets, with the test accuracy reported based on the epoch that achieves the highest validation accuracy. We search model hyperparameters including walk length of random walk, hidden dimension, and dropout. The results of HOtrans are averaged over 10 runs with random weight initializations.

## A.6 MORE RESULTS.

**Effect of position encoding.** Based on Spectral Clustering, we test the role of positional encoding for the proposed HOTtrans. We compare two popular positional encoding methods including Laplacian-based (lpe) and random walk positional encoding (rwpe) to HOtrans without any positional encoding. From the table, the gap in performance is minor between the two positional encoding methods over all datasets. While without positional encoding, HOtrans achieves better performance on heterophilic datasets, such as, Cornell, Texas, and Wisconsin. This is because the positional encoding methods (such as Laplacian PE) usually encode the original graph connections, thus, integrating positional encoding will lead to a negative effect for these heterophilic datasets which contain massive noisy information in graph structure.

