# OpenReview forum: "Exploring High-Order Message-Passing in Graph Transformers"
_ICLR.cc/2024/Conference — Submitted to ICLR 2024_

### Official Review · Reviewer_eGUc · 2023-10-27

**Soundness:** 2 fair
**Presentation:** 3 good
**Contribution:** 2 fair
**Rating:** 3
**Confidence:** 5

**Summary:**

This paper proposes a new graph representation learning method called HOtrans, which builds upon the Transformer architecture. HOtrans introduces a three-step message-passing approach to capture high-order correlations and long-range dependent information in graphs. Specifically, it first constructs communities, and then involves message passing between node to virtual node, virtual node to virtual node, and virtual node to a node. Experiments results showed the effectiveness on Homophilic and Heterophilic graphs.

**Strengths:**

1.  The proposed HOtrans can utilize high order information, and serves as a general framework for many existing graph transformers.

2. HOtrans achieves a competitive performance.

3. This paper is clearly presented and easy to follow.

**Weaknesses:**

1. The key components of HOtrans are basically built upon existing concepts, e.g., virtual node, community sampling etc, which makes this paper less disruptive.

2 . High order information is important and existing solutions usually adopt hypergraph method. The proposed HOtrans resembles hypergraph convolutional networks, and there exist many hypergraph solutions, but the authors didn’t discuss the differences, without making compasions in the experiments.

3. I think the community sampling method will have a big impact on the model performance and should be datasets dependent. However, this paper is mainly based on existing random walk and sepctal clustering.

minor:

1. I cannot find the corresponding text for table 3. What is this table for?

2. Typos: A.1, gapformer should be HOtrans?  Also, I suggest the authors to have a table summarizing the complexity compasions with existing models.

**Questions:**

1. How the proposed method compared with hypergraph approaches?

2. Can the proposed HOtrans be applied to graph level tasks and how about the performance?

---

> ### Author Response · Authors · 2023-11-23
> **Response to Reviewer eGUc**
>
> Thank you for your comments and suggestions, and they are exceedingly helpful for us to improve our paper. In the following, your comments are first stated and then followed by our point-by-point responses.
>
> **Q1.** The key components of HOtrans are basically built upon existing concepts, e.g., virtual node, community sampling etc, which makes this paper less disruptive.
>
> **A1.** HOtrans is proposed to capture local, high-order, and global information in the graph with a three-step message-passing scheme. The proposed three-step message-passing framework propagates and aggregates information based on extracted communities, specifically achieving local and high-order information in the community and obtaining global information among communities which act as bridges. The proposed new message-passing architecture unifies message-passing and Graph Transformer.
>
> Regarding detailed design, our aim is not to propose a new sampling technique. From the experimental results, we can observe the effectiveness of using existing sampling methods, including spectral clustering and random walk sampling. The introduced virtual nodes in HOtrans act as bridges to connect the entire graph, playing different roles compared to other methods.
>
> **Q2.** High order information is important and existing solutions usually adopt hypergraph method. The proposed HOtrans resembles hypergraph convolutional networks, and there exist many hypergraph solutions, but the authors didn’t discuss the differences, without making compasions in the experiments. / How the proposed method compared with hypergraph approaches?
>
> **A2.** Theoretically, we compare the proposed HOtrans with hypergraph convolutional networks in A.3. As we analyzed in A.3, dropping out the second step (V2V-ATTN), our HOtrans is similar to popular hypergraph-based neural networks from the perspective of message-passing.
> Empirically, in subsection 5.1, we compare HOtrans with popular hypergraph convolutional networks HGNN. HOtrans improves upon the state-of-that art GNN method-HGNN by a margin of 1.3% on Cora.
>
> **Q3.** I think the community sampling method will have a big impact on the model performance and should be datasets dependent. However, this paper is mainly based on existing random walk and sepctal clustering.
>
> **A3.** From the experimental results, we can see the effectiveness of using existing sampling methods including spectral clustering and random walk sampling. We will further evaluate the performance of HOtrans by using other sampling methods.
>
> **Q4.** I cannot find the corresponding text for table 3. What is this table for?
>
> **A4.** Thanks for carefully read the paper. We perform a series of ablation studies to test the importance of some designs in our proposed HOtrans. We report the results in Table 3.
> We will carefully check the whole paper in the final version.
>
> **Q5.** Typos: A.1, gapformer should be HOtrans? Also, I suggest the authors to have a table summarizing the complexity compasions with existing models.
>
> **A5.** Thanks for carefully read the paper. We will carefully check the whole paper in the final version, and add a table to summarize the complexity of existing models.
>
> **Q6.** Can the proposed HOtrans be applied to graph level tasks and how about the performance?
>
> **A6.** As a general graph representation learning method, HOtrans can be applied for graph level tasks. We will further evaluate the performance of HOtrans for graph classification task.

---

### Official Review · Reviewer_7DnL · 2023-10-30

**Soundness:** 3 good
**Presentation:** 2 fair
**Contribution:** 2 fair
**Rating:** 5
**Confidence:** 4

**Summary:**

The paper introduces a Transformer architecture called HOtrans, which employs a higher-order message passing strategy. The model initially extracts communities, each centered around a virtual node, from the entire graph. Subsequently, the proposed attention mechanism comprises three steps: first, information is aggregated towards higher-order virtual nodes; second, information is exchanged among representations of these higher-order virtual nodes; and finally, the updated information is relayed back to the original graph nodes. In the experimental study, the authors demonstrate the effectiveness of HOtrans in node classification tasks.

**Strengths:**

1. The proposed method not only delivers competitive performance on homophilic datasets but also surpasses the state-of-the-art methods on heterophilic datasets.

2. The proposed HOtrans is scalable to large graphs.

**Weaknesses:**

1. The novelty of the paper appears somewhat limited. Both community sampling techniques and the virtual node design have been previously introduced in other studies. This work seems to primarily amalgamate these existing strategies.

2. The proposed method may not be competitive to some linear graph transformers. For example, NodeFormer [1], with less computational resource and fewer training samples, gets better results on full attention graphs, e.g., on Cora with 88.80% accuracy versus 88.11% obtained by the proposed method.

3. It is claimed that the proposed method can capture long-range dependency in graphs. Results on some long-range graph datasets are expected, e.g., PascalVOC-SP from LRGB [2].

4. It would be interesting to try learning-based community sampling methods to assess its effect to the proposed model.

[1] Wu Q, Zhao W, Li Z, Wipf D.P, Yan J. Nodeformer: A scalable graph structure learning transformer for node classification. Advances in Neural Information Processing Systems. 2022 Dec 6;35:27387-401.

[2] Dwivedi V.P, Rampášek L, Galkin M, Parviz A, Wolf G, Luu A.T, Beaini D. Long range graph benchmark. Advances in Neural Information Processing Systems. 2022 Dec 6;35:22326-40.

Typo: In appendix A.1, it should be “complexity of HOtrans”, instead of “complexity of Gapformer”.

Minor problem: The citations for Gapformer in Tables 1 and 2 are missing.

**Questions:**

Please refer to the weaknesses.

---

> ### Author Response · Authors · 2023-11-23
> **Response to Reviewer 7DnL**
>
> Thank you for your comments and suggestions, and they are exceedingly helpful for us to improve our paper. In the following, your comments are first stated and then followed by our point-by-point responses.
>
> **Q1.** The novelty of the paper appears somewhat limited. Both community sampling techniques and the virtual node design have been previously introduced in other studies. This work seems to primarily amalgamate these existing strategies.
>
> **A1.** HOtrans is proposed to capture local, high-order, and global information in the graph with a three-step message-passing scheme. The proposed three-step message-passing framework propagates and aggregates information based on extracted communities, specifically achieving local and high-order information in the community and obtaining global information among communities which act as bridges. The proposed new message-passing architecture unifies message-passing and Graph Transformer.
>
> Regarding detailed design, our aim is not to propose a new sampling technique. From the experimental results, we can observe the effectiveness of using existing sampling methods, including spectral clustering and random walk sampling. The introduced virtual nodes in HOtrans act as bridges to connect the entire graph, playing different roles compared to other methods.
>
> **Q2.** The proposed method may not be competitive to some linear graph transformers. For example, NodeFormer [1], with less computational resource and fewer training samples, gets better results on full attention graphs, e.g., on Cora with 88.80% accuracy versus 88.11% obtained by the proposed method.
>
> **A2.** Using the same dataset split in HOtrans, we achieve 86.00% accuracy on 10 runs on Cora for NodeFormer. We can see that HOtrans can achieve better performance (88.11%) compared to NodeFormer.
>
> **Q3.** It is claimed that the proposed method can capture long-range dependency in graphs. Results on some long-range graph datasets are expected, e.g., PascalVOC-SP from LRGB [2].
>
> **A3.** We will try to evaluate HOtrans on some long-range graph datasets.
>
> **Q4.** It would be interesting to try learning-based community sampling methods to assess its effect to the proposed model.
>
> **A4.** Thanks for your suggestion. We considered the learning-based methods, but the learning-based sampling methods suffer from computation complexity, and cannot achieve better performance than non-learning methods as analyzed in PatchGT [1]. We will try other sampling methods to further evaluate HOtrans.

---

### Official Review · Reviewer_e8WT · 2023-10-31

**Soundness:** 2 fair
**Presentation:** 3 good
**Contribution:** 2 fair
**Rating:** 3
**Confidence:** 4

**Summary:**

In this work, the authors propose a new architecture for graph learning, HOtrans, specializing in transductive node classification tasks. The approach involves three steps: (a) community detection, (b) self-attention between community representations, and (c) update of the node representations from community representations. The authors compare their model on various node classification datasets, including both homophilic and heterophilic graphs.

**Strengths:**

**Strengths:**
- The paper addresses an important problem.
- The proposed model compares well against most of the presented baselines
- The empirical study is thorough, and the paper presents ablation studies to gain further understanding of the essential components of the proposed architecture.

**Weaknesses:**

**Weaknesses** (in the order of importance):
- The most important issue with this work is that the empirical results of the proposed model are not very convincing. Overall, the presented improvements of HOtrans are well within the standard deviation of prior works such as Graphormer [1]. The only exception to this are Texas and Wisconsin, where HOtrans is significantly better than previous models. On Actor, the authors also improve significantly over Gapformer but are not significantly better than a standard transformer with Laplacian encodings [2]. Considering that HOtrans is overall very close in performance to Gapformer and the fact that Gapformer also explores applying self-attention to pooled representations of sets of nodes, it is not convincing that HOtrans adds much value over Gapformer.

- The motivation of HOtrans is quite weak. The authors mention that HOtrans was developed to capture long-range dependencies and to aggregate the information of sets of nodes instead of single nodes. However, such approaches have already been explored, e.g., with the Graph-ViT [3]. Further, the ability of graph transformers to capture long-range dependencies has already been demonstrated in e.g., GraphGPS [4] and even in the context of transductive node classification [2]. It would be helpful if the authors could empirically compare their approach to these prior works.

- The datasets used for heterophilic node classification, in particular Cornell, Texas, and Wisconsin, have recently been shown to have substantial weaknesses, such as their very small size and imbalanced classes [5]. The claim that HOtrans performs favorably to graph transformers on heterophilic datasets could be further substantiated by additional results on more suitable datasets such as the ones in [5]. On Actor, which is the only heterophilic dataset that does not suffer from the aforementioned issues, HOtrans performs no better than a standard transformer with Laplacian encodings [2] (as already mentioned above).

In summary, while the proposed model, HOtrans, outperforms most presented baselines, it only outperforms the best prior method on each dataset on 2/10 datasets. Most crucially, the empirical results are very close to those of Gapformer on nearly all datasets and it is not made clear how the proposed method is otherwise favorable to Gapformer.

References:
[1]: Gapformer: Graph Transformer with Graph Pooling for Node Classification, Liu et al. 2023
[2]: Attending to Graph Transformers, Müller et al. 2023
[3]: A Generalization of ViT/MLP-Mixer to Graphs, He et al. 2022
[4]: Recipe for a General, Powerful, Scalable Graph Transformer, Rampasek et al. 2022
[5]: A Critical Look at the Evaluation of GNNs under Heterophily: Are We Really Making Progress, Platanov et al., 2023

**Questions:**

What added value does HOTrans provide compared to Gapformer considering that both models perform nearly the same across all datasets?

How does HOtrans perform to prior graph transformers that aggregate sets of nodes (e.g., Graph-ViT) or have demonstrated to capture long-range dependencies (e.g., GraphGPS)?

---

> ### Author Response · Authors · 2023-11-23
> **Response to Reviewer e8WT**
>
> Thank you for your comments and suggestions, and they are exceedingly helpful for us to improve our paper. In the following, your comments are first stated and then followed by our point-by-point responses.
>
> **Q1.** The most important issue with this work is that the empirical results of the proposed model are not very convincing. Overall, the presented improvements of HOtrans are well within the standard deviation of prior works such as Graphormer [1]. The only exception to this are Texas and Wisconsin, where HOtrans is significantly better than previous models. On Actor, the authors also improve significantly over Gapformer but are not significantly better than a standard transformer with Laplacian encodings [2]. Considering that HOtrans is overall very close in performance to Gapformer and the fact that Gapformer also explores applying self-attention to pooled representations of sets of nodes, it is not convincing that HOtrans adds much value over Gapformer.
>
> **A1.** The improvements of HOtrans over Graphormer are evident across all datasets, for example, achieving 88.11% compared to 67.71% on Cora.
>
> In existing Graph Transformer architectures, Gapformer, which aggregates global information through pooling methods, also demonstrates excellent performance on homophilic datasets like Cora. Theoretically, Gapformer can be considered an equivalent substitute for our HOtrans in case where the entire graph is treated as a community. However, the significant advantage of HOtrans over Gapformer lies in encoding high-order representations of graphs; in other words, HOtrans can be applied to both graphs and hypergraphs, whereas Gapformer is designed for graphs.
>
> HOtrans achieves the best performance across all heterophilic datasets and the majority of homophilic datasets. However, for homophilic datasets such as Cora and Citeseer, where local information holds more importance, GNN-based methods like APPNP outperform Gapformer. Based on Transformer architecture, HOtrans outperforms APPNP on Cora and Citeseer by considering local, high-order, and long-range dependency information. Furthermore, leveraging other suitable sampling methods for community extraction has the potential to further enhance HOtrans.
>
> **Q2.** The motivation of HOtrans is quite weak. The authors mention that HOtrans was developed to capture long-range dependencies and to aggregate the information of sets of nodes instead of single nodes. However, such approaches have already been explored, e.g., with the Graph-ViT [3]. Further, the ability of graph transformers to capture long-range dependencies has already been demonstrated in e.g., GraphGPS [4] and even in the context of transductive node classification [2]. It would be helpful if the authors could empirically compare their approach to these prior works. / How does HOtrans perform to prior graph transformers that aggregate sets of nodes (e.g., Graph-ViT) or have demonstrated to capture long-range dependencies (e.g., GraphGPS)?
>
> **A2.** Graph-ViT [3] is proposed as an alternative approach to capture long-range dependencies by leveraging ViT/MLP-Mixer in computer vision for graph classification, while GraphGPS is also proposed for graph classification. We will try to compare them for graph classification task.
>
> **Q3.** The datasets used for heterophilic node classification, in particular Cornell, Texas, and Wisconsin, have recently been shown to have substantial weaknesses, such as their very small size and imbalanced classes [5]. The claim that HOtrans performs favorably to graph transformers on heterophilic datasets could be further substantiated by additional results on more suitable datasets such as the ones in [5]. On Actor, which is the only heterophilic dataset that does not suffer from the aforementioned issues, HOtrans performs no better than a standard transformer with Laplacian encodings [2] (as already mentioned above).
>
> **A3.** We will try to add more experimental results on other heterophilic datasets.
>
> **Q4.** What added value does HOTrans provide compared to Gapformer considering that both models perform nearly the same across all datasets?
>
> **A4. ** We unify the message-passing and Graph Transformer by the proposed HOtrans framework. Theoretically, Gapformer can be viewed as a special case of the proposed HOtrans. Please see the analysis from section4. More empirical analysis can be found in **A1**.

---

> > ### Comment · Reviewer_e8WT · 2023-12-01
> >
> > Thank you for the rebuttal, the authors did not provide the requested additional experimental results, so I will not keep my score.

---

### Official Review · Reviewer_akHZ · 2023-10-31

**Soundness:** 3 good
**Presentation:** 4 excellent
**Contribution:** 3 good
**Rating:** 5
**Confidence:** 5

**Summary:**

This work proposes a virtual node based graph transformer to capture the high-order structure information of the graph. The designed approach is by (1) node clustering by ClusterGCN or random walk-based GraphSaint, (2) attentions at three levels including node-to-virtual-node, virtual-node-to-virtual-node, and virtual-node-to-node. This work is evaluated by node classification tasks on both homophily and heterophily graphs, which show better or competitive performance over the state-of-the-art baselines.

**Strengths:**

1. This work introduces an interesting approach to leveraging virtual nodes to improve the capability of graph transformers to capture graph high-order information, for node-level tasks. Prior works leverage virtual nodes to solve graph-level tasks (e.g., graph classification).

2. The designed approach is to some extent generic and can be specified into different types of GNNs according to how communities/clusters are defined.

3. The proposed method indeed performs better than the state-of-the-art on most datasets, except ogbn-arxiv dataset. Ablation studies are done to show the effectiveness of different components.

**Weaknesses:**

1. It is not clear how theorem 4.1 can demonstrate the power of the designed method from the theoretical perspective. In other words, given that the proof of Theorem 4.1 is based on top of Proposition 4.1 while there's no approximation error provided, it is vague to define `arbitrarily well' mentioned in Theorem 4.1.

2. The explanations of why positional encoding hurts the performance are a bit insufficient. I agree with the authors that Laplacian PE can hurt the performance on heterophily datasets since heterophily graph usually requires non-low-pass filters for aggregations. However, it's not clear why random walk based encoding also performs worse. Besides, both encoding methods hurt the performance on both homophily and heterophily datasets.

3. Eq. 3 seems to be incorrect. \bar{x}_i is defined as a zero vector, as described in the paper, so \bar{q}_i is essentially just a zero vector and thus carries no information at all.

**Questions:**

1. How do you define 'arbitrarily well' in Theorem 4.1 mathematically?
2. Can you elaborate more about the reasons why positional encodings perform generally worse than not using positional encodings on both homophily and heterophily datasets?
3. Why is \bar{x}_i defined as a zero vector?

---

> ### Author Response · Authors · 2023-11-23
> **Response to Reviewer akHZ**
>
> Thank you for your comments and suggestions, and they are exceedingly helpful for us to improve our paper. In the following, your comments are first stated and then followed by our point-by-point responses.
>
> **Q1.** It is not clear how theorem 4.1 can demonstrate the power of the designed method from the theoretical perspective. In other words, given that the proof of Theorem 4.1 is based on top of Proposition 4.1 while there's no approximation error provided, it is vague to define `arbitrarily well' mentioned in Theorem 4.1.
>
> **A1.** The demonstration of the proposed framework is powerful as the self-attention of vanilla Graph Transformers in the general case can be found in Appendix A.4.
>
> **Q2.** The explanations of why positional encoding hurts the performance are a bit insufficient. I agree with the authors that Laplacian PE can hurt the performance on heterophily datasets since heterophily graph usually requires non-low-pass filters for aggregations. However, it's not clear why random walk based encoding also performs worse. Besides, both encoding methods hurt the performance on both homophily and heterophily datasets.
>
> **A2.**  The positional encoding (PE) is employed as input features to capture the positional or structural information of a graph in Transformer. In contrast, our proposed framework has already incorporated these types of information through community sampling and a specially designed three-step message-passing scheme. Specifically, the adopted random walk sampling or spectral sampling methods take into account the local or global structural information in the graph. The structural information is then implicitly encoded through propagation and aggregation based on the community. Consequently, positional encoding does not provide significant benefits for our proposed method. This is evident from Table 5 in the Appendix, where the performance differences with or without positional encoding methods on Cora or Citeseer are minor.
>
> For heterophilic datasets, as the reviewer agreed that since heterophily graph usually requires non-low-pass filters for aggregations, thus Laplacian PE can hurt the performance. Considering that Random Walk PE encodes the local connections between nodes, it also fails to improve performance. Moreover, we maintain a fixed walk length for random walk embedding for all datasets; using hyperparameter selection to choose suitable parameters for different datasets will mitigate its impact.
>
> **Q3.** Eq. 3 seems to be incorrect. \bar{x}_i is defined as a zero vector, as described in the paper, so \bar{q}_i is essentially just a zero vector and thus carries no information at all.
>
> **A3.** Thanks for your question. The $\bar{q}_i$ carries no information at the beginning but it aggregates the information from the community after the Eq. 3.

---

### Meta-Review · Area_Chair_FiNS · 2023-12-05

**Metareview:**

This paper introduces HOtrans, a graph Transformer architecture designed for transductive node classification tasks. The method involves three key steps: community detection, self-attention between community representations, and updating node representations from these community representations. HOtrans demonstrates improved or competitive performance in node classification on both homophily and heterophily graphs, showcasing its ability to capture high-order structure information effectively.

While the proposed HOtrans model shows promising results, the reviewers have identified several weaknesses that need to be addressed:

1. The paper's novelty is limited, as it mainly combines existing community sampling techniques and virtual node design.
2. The motivation of the paper is unclear.

Based on these weaknesses, we recommend rejecting this paper. We hope this feedback helps the authors improve their paper.

**Justification For Why Not Higher Score:**

The reviewers unanimously believe the paper should be rejected.

**Justification For Why Not Lower Score:**

N/A

---

### Decision · Program_Chairs · 2024-01-16

Reject